# The Influence of Diamond Burnishing Process Parameters on Surface Roughness of Low-Alloyed Aluminium Workpieces

Viktoria Ferencsik and Gyula Varga * 

Institute of Manufacturing Science, University of Miskolc, H-3515 Miskolc, Hungary;
ferencsik.viktoria@uni-miskolc.hu
* Correspondence: gyula.varga@uni-miskolc.hu

**Abstract:** This study describes the determination and optimization of burnishing process parameters and their effects on surface roughness of EN AW-2011 aluminium alloy workpieces. The process has a low environmental load and the chip-free burnishing process improves the integrity of the machined surface, but to achieve this, the different burnishing parameters, for example, burnishing force, feed rate, speed and number of passes, must be properly defined according to the material of the workpiece. In our research, a full factorial experimental design method is used to plan and carry out the experiments and to determine the most appropriate parameter range for this material quality.

**Keywords:** cold plastic forming; burnishing; surface roughness; full factorial experimental design



## 1. Introduction

The appearance of new material qualities has led to the rise of new sectors and processes, and nowadays, during the fourth industrial revolution, a prominent role has been given to guaranteeing the best possible surface quality of manufactured parts [1]. Finishing operations have been developed that can productively provide ever-improving macro- and micro-geometric accuracy since different engineering industries have realized that the final dimensions, corrosion resistance and, in fact, proper operation of a machining part depends on the surface roughness [2–4], among other factors. In recent decades, more emphasis has been placed on non-chip removal technologies such as the cold plastic surface hardening processes. Surface burnishing technology is used as an effective method for reducing surface roughness, increasing the micro-hardness of the sub-surface layer, and improving the shape correctness, while not requiring large amounts of coolant, so that overall low environmental impact and economical machining can be realized [5–7]. This process eliminates the disadvantages (tool marks, scratches) that are associated with the application of conventional chip removal techniques like turning or grinding, thereby preventing dissipation and surface damage [8].

Many researchers have experimentally shown that mechanical surface treatment increases surface wear resistance and surface integrity [9–11] and it also has been shown that these procedures increase the corrosion resistance of the treated surfaces [12]. Dzionk et al. [13] demonstrated that burnishing greatly improves the quality of the hardened turned surface by examining various 2D and 3D roughness parameters. Special care was taken to ensure that the values of the feed rates of hard turning and burnishing did not coincide; this is important so that the flattening of the peaks can take place sufficiently. They observed that the deformation process of surface irregularities occurs primarily in the zone of peaks.

Sharma and Kapor [14] developed a new type of burnishing tool and experiments were carried out according to the Taguchi L9 orthogonal array. They found that burnishing speed and number of passes have major influence on determining the surface roughness and hardness, but interestingly, the feed rate proved to be an insignificant parameter. Ghodake

et al. [15] also came to this conclusion according to their analytical and experimental investigations, which were based on full factorial experimental design and statistical analysis using ANOVA and the regression method. They established that the percentage contribution of different burnishing parameters which show depth of penetration is greatest (95.8623%) for the change of surface roughness, while feed rate affects it only 2.522%.

In contrast, for example Basak et al. [7] identified a specific range within which the feed rate is the most advantageous for improving roughness. Multiple regression and ANOVA were applied also for examining different parameters. According to the compared results, the influence of the number of revolutions (so burnishing speed) and the ball diameter were not as great as the burnishing force and feed rate. However, the combined effect of these two factors was not examined.

Randjelovic et al. [16] focused on examining a single parameter: the depth of penetration. Kinematically compatible FEM simulations, then experimental investigations, were performed and, according to both results, they determined that the minimal surface roughness can be achieved by setting ball penetration depth close to the maximum peak height. The explanation is most likely that this condition is the optimum for material flow [16]. Ferencsik and Gál [17] also highlighted one parameter from the finite element examination, the burnishing force. According to their DEFORM 2D simulation model, the degree of average surface roughness decreases in direct proportion to the increase of the force during burnishing of low-alloyed aluminium. In this work, the surface points of the workpiece were determined with the physical measurement points that were imported to the DEFORM FE code [17]. Banh et al. [18] also dealt with the numerical simulation of the effect of burnishing force on surface roughness, but they described the pre-burnishing surface roughness with parabolic approximation. This made the study more complicated, but the measured and simulated values still show a good approximation, so their model can be applied correctly.

Dzierwa and Markopoulos [19] investigated the effect of burnishing input parameters on surface topography, residual stress and tribological properties of 42CrMo4-hardened steel, when the pre-treatment process was grinding. Based on the results of the comprehensive experiment, burnishing pressure force was found to be the most influential parameter, while burnishing speed and stepover had negligible effects. Kovács et al. examined the effect of magnetic-assisted ball burnishing on C45 steel [20]. They investigated the change of corrosion resistance when changing the burnishing parameters. The corrosion rate and the related technological parameters were optimized by the Taguchi method. The burnishing forms an invisible so-called "light coating" on the surface of the workpiece. In this respect, it can be related to the study by Řehoř et al. [21], which deals with the evaluation of the surface quality of a wear-resistant hard coating. Surface-roughness testing is essential to determine the predictability of both light and hard coatings.

In this investigation, we deal with the effect of burnishing force (F), feed rate (f), speed (v) and number of passes (i) on average surface roughness examining the correlation between the parameters on EN AW-2011 external cylindrical workpieces. The diamond-burnishing process was performed on an OPTIMUM type OPTIturn S600 CNC lathe after the turning pre-treatment operation.

## 2. Materials and Methods

In the mechanical engineering industry, mainly grinding is used to improve the surface quality of metals and their alloys [22], but more and more industrial sectors (automotive, aerospace, chemical, food, etc.) use non-ferrous materials due to their low density and favourable mechanical properties [7,23–25]. However, abrasive machining of these materials is difficult due to the high heat generation; in this case, surface burnishing can be used as an efficient and advantageous solution, as it can be performed below the recrystallization temperature.

The chemical composition of the EN AW-2011-type aluminium workpiece material can be seen in Table 1.

**Table 1.** Chemical composition of the tested aluminium alloy (% by volume) [26].

|  | Si | Fe | Cu | Mn | Mg | Cr | Zn | Bi | Pb | Al |
|---|---|---|---|---|---|---|---|---|---|---|
| Min |  |  | 3.3 | 0.5 | 0.4 |  |  | 0.3 | 0.2 | 89.3 |
| Max | 0.4 | 0.8 | 4.6 | 1 | 1.8 | 0.1 | 0.2 | 0.6 | 0.6 | 95.1 |

Burnishing can be performed without the use of coolants. Only the use of lubricant is necessary, as in other environmentally friendly manufacturing systems [27,28]. Burnishing is a distinguished process to improve surface properties, in which the profile irregularities will be formed under the effect of burnishing force by tool based on the principle of sliding friction. No chips, sparks or dust are generated during machining; in addition, the need for coolant is minimal and can sometimes be omitted, so we can implement environmentally friendly and cost-effective machining [29–32]. This is preferred for finishing of hydraulic and other cylinders, pistons, bearing housings, bushings and pins [9,22]. Its field of application generally includes the automotive, aeronautics and aerospace industries [33,34].

Burnishing of external cylindrical surfaces can be done on universal or CNC lathes, and Figure 1 (on the basis of [35,36]) shows how the required compressive force, which must exceed the flow limit of the workpiece material, is generated by a spring integrated in the tool, and then the tool is pulled along the surface of the rotating workpiece at a feed rate (f).

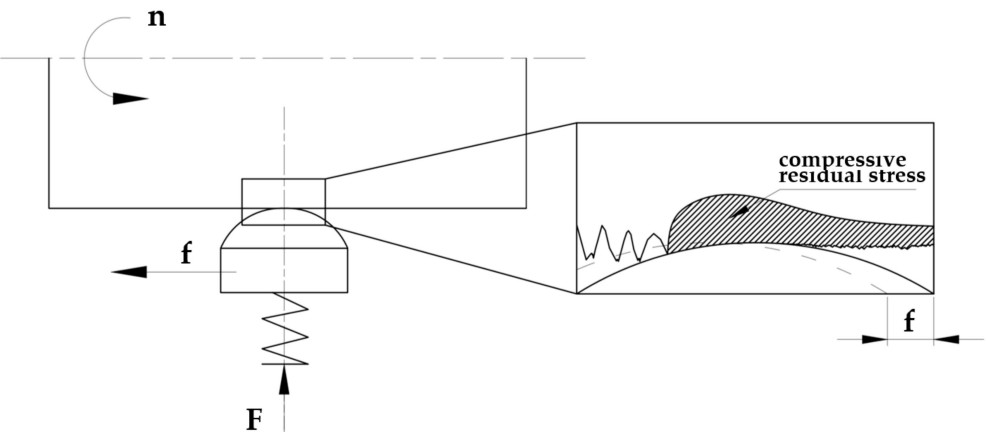

**Figure 1.** Schematic diagram of sliding friction burnishing (F—burnishing force; n—revolution; f—feed rate).

Our burnishing experiments were executed with an OPTIMUM type OPTIturn S600 CNC lathe with a PCD spherical burnishing tool (r = 3.5 mm) and the kinematic viscosity of the applied manual dosing oil was $\nu = 70$ mm$^2$/s. Burnishing operations were preceded by a finishing turning set at $f_{1t} = 0.2$ and then $f_{2t} = 0.15$ mm/rev feed rate.

The determination of the burnishing conditions is based on approximate calculations, the existing experimental results of different burnished materials with identical or similar properties, and available universal nomograms and special standards. However, the large number of parameters and their interactions with each other make this process complicated, which can only be solved by performing a large number of lengthy experiments. To avoid this, a full factorial experimental design method is used, which allows several factors to be examined simultaneously. To reduce the number of experiments, the number of settings tested is usually maximized to two per factor. This value is sufficient to detect the significance of the factors and, in some cases, to determine the optimal setting range. Factorial designs are simple and logical to handle, which is why they can be easily used in industrial practice [37–39].

Table 2 summarizes all the examined burnishing parameters as factors in two levels for two experiments.

**Table 2.** Burnishing parameters.

| No. | Burnishing Parameters I | | | Burnishing Parameters II | | | Transformed Parameters | | |
|---|---|---|---|---|---|---|---|---|---|
| | F (N) | f (mm/rev) | i (ø) | F (N) | f (mm/rev) | v (m/min) | $x_1$ | $x_2$ | $x_3$ |
| 1 | 10 | 0.001 | 1 | 10 | 0.001 | 15 | −1 | −1 | −1 |
| 2 | 20 | 0.001 | 1 | 20 | 0.001 | 15 | +1 | −1 | −1 |
| 3 | 10 | 0.005 | 1 | 10 | 0.005 | 15 | −1 | +1 | −1 |
| 4 | 20 | 0.005 | 1 | 20 | 0.005 | 15 | +1 | +1 | −1 |
| 5 | 10 | 0.001 | 3 | 10 | 0.001 | 30 | −1 | −1 | +1 |
| 6 | 20 | 0.001 | 3 | 20 | 0.001 | 30 | +1 | −1 | +1 |
| 7 | 10 | 0.005 | 3 | 10 | 0.005 | 30 | −1 | +1 | +1 |
| 8 | 20 | 0.005 | 3 | 20 | 0.005 | 30 | +1 | +1 | +1 |

In order to make the change in surface roughness caused by burnishing even more illustrative, a dimensionless ratio was created based on this equation according to E-Taweel and El-Axir [40]:

$$\Delta\rho_{R_x}\% = \left( \frac{R_{x\text{turned}} - R_{x\text{burnished}}}{R_{x\text{turned}}} \right) \times 100\%, \tag{1}$$

where

$R_{x\text{ turned}}$      Surface-roughness parameter remaining after turning,
$R_{x\text{ burnished}}$      Surface-roughness parameter remaining after burnishing,
$\Delta\rho_{R_x}\%$      Percentage value of the improvement ratio.

The higher the value of $\Delta\rho_{R_x}\%$, the greater the improvement due to burnishing.

Due to the macroscopic and microscopic surface defects, a real surface—the machined part of the material that is often affected by a variety of processes—is different from the ideal or quasi-ideal surface. Surface roughness means the microscopic geometric features of peaks and valleys on a machined part, and it has a great effect on corrosion resistance, wear resistance, sealing, contact stiffness and fatigue strength [41–44]. This is one of the reasons the analysis of surface roughness has been developed for more than 100 years and measurement methods have been widely used in industrial practice.

In our investigations, the different roughness parameters were investigated on an Altisurf © 520 (Altimet SAS, Sainte-Helene-du-Lac, France) measuring device with CL2 confocal chromatic sensor (Figure 2) before and after burnishing in three different positions rotated by 120°. For the evaluation of the data, a cut-off of λc = 0.8 mm and Gauss filter were applied. The advantage of using this machine is that the software can not only perform a particular measurement, but it can also be done in one setting in succession, so a pre-programmed measurement process can be carried out. This means that multiple measurements can be defined one after the other or axis movements can be set.

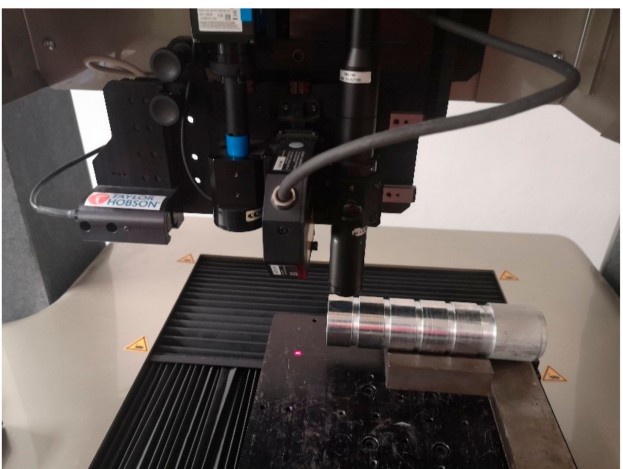

**Figure 2.** Working area of the Altisurf 520 measuring machine with CL2 confocal chromatic sensor.

From the many parameters that characterize the surface roughness, we tried to select those that best reflect the changes caused by the burnishing process and that are also widely used in engineering practice. The names, definitions and calculation formulas are summarized in Table 3 [45]. The valid standard of geometrical product specifications (GPS) is ISO 22081:2021 [46].

**Table 3.** The examined surface-roughness parameters.

| Mark | Name | Definition | Formula |
|---|---|---|---|
| $R_a$ | Average roughness | Arithmetic means of the absolute height of the profile | $\frac{1}{L}\int_0^L |y|\,dx$ |
| $R_q$ | Root mean square roughness | Root mean square of the height of the profile | $\sqrt{\frac{1}{L}\int_0^L y^2\,dx}$ |
| $R_z$ | Average roughness height | Average absolute value of the five highest peaks and the five lowest valleys | $\frac{\sum y_{P_i} + \sum y_{v_i}}{5}$ |
| $R_t$ | Maximum height of the profile | Total height of the assessed profile | $\max_i y_i - \min_i y_i$ |

## 3. Results

The measured and calculated values of the different roughness parameters for both experimental variations are summarized in Tables 4–7.

**Table 4.** The measured values and the calculated improvement ratios for $R_a$.

| No. | $R_a$ (µm) I. | | $\Delta\sigma_{Ra}$ (%) | $R_a$ (µm) II. | | $\Delta\sigma_{Ra}$ (%) |
|---|---|---|---|---|---|---|
| | Turned | Burnished | | Turned | Burnished | |
| 1 | 1.2260 | 0.3457 | 71.80 | 1.0117 | 0.4231 | 58.18 |
| 2 | 0.9213 | 1.2686 | −37.69 | 0.9299 | 0.2631 | 71.71 |
| 3 | 0.9947 | 0.3599 | 63.82 | 0.9374 | 0.3040 | 67.57 |
| 4 | 1.0679 | 0.5875 | 44.99 | 0.8834 | 0.3017 | 65.85 |
| 5 | 1.0118 | 1.8215 | −80.06 | 0.9524 | 0.4891 | 48.65 |
| 6 | 1.0622 | 2.2249 | −109.46 | 1.1319 | 0.4141 | 63.42 |
| 7 | 0.9450 | 0.2516 | 73.38 | 1.0559 | 0.6421 | 39.19 |
| 8 | 1.0741 | 1.3817 | −28.64 | 0.9814 | 1.2703 | −29.44 |

**Table 5.** The measured values and the calculated improvement ratios for $R_q$.

| No. | $R_q$ (µm) I. | | $\Delta\sigma_{Rq}$ (%) | $R_q$ (µm) II. | | $\Delta\sigma_{Rq}$ (%) |
|---|---|---|---|---|---|---|
| | Turned | Burnished | | Turned | Burnished | |
| 1 | 1.4181 | 0.5772 | 59.29 | 1.2535 | 0.5190 | 58.59 |
| 2 | 1.1423 | 1.5577 | −36.37 | 1.1432 | 0.3272 | 71.38 |
| 3 | 1.2361 | 0.4491 | 63.67 | 1.1654 | 0.3826 | 67.17 |
| 4 | 1.3050 | 0.7527 | 42.32 | 1.1190 | 0.3697 | 66.96 |
| 5 | 1.2184 | 2.3315 | −91.36 | 1.1799 | 0.6170 | 47.71 |
| 6 | 1.2934 | 2.7763 | −114.65 | 1.3682 | 0.5019 | 63.32 |
| 7 | 1.1659 | 0.3142 | 73.05 | 1.2873 | 0.7910 | 38.55 |
| 8 | 1.3002 | 1.6946 | −30.33 | 1.1968 | 1.5383 | −28.53 |

**Table 6.** The measured values and the calculated improvement ratios for $R_z$.

| No. | $R_z$ (µm) I. | | $\Delta\sigma_{Rz}$ (%) | $R_z$ (µm) II. | | $\Delta\sigma_{Rz}$ (%) |
|---|---|---|---|---|---|---|
| | Turned | Burnished | | Turned | Burnished | |
| 1 | 6.0651 | 2.7207 | 55.14 | 6.1135 | 2.7146 | 55.59 |
| 2 | 5.6889 | 6.7087 | −17.93 | 5.7399 | 1.8995 | 66.91 |
| 3 | 6.0100 | 2.5422 | 57.70 | 6.3137 | 2.3293 | 63.11 |
| 4 | 5.9449 | 4.0704 | 31.53 | 6.5407 | 2.0272 | 69.01 |
| 5 | 5.8470 | 8.9717 | −52.54 | 5.9786 | 2.9443 | 50.75 |
| 6 | 6.1230 | 10.3915 | −69.71 | 6.4803 | 2.4865 | 61.63 |
| 7 | 5.9639 | 1.9799 | 66.80 | 6.5035 | 4.0826 | 37.22 |
| 8 | 5.6166 | 6.8064 | −21.18 | 5.8492 | 7.0841 | −21.11 |

**Table 7.** The measured values and the calculated improvement ratios for $R_t$.

| No. | $R_t$ (µm) I. | | $\Delta\sigma_{Rt}$ (%) | $R_t$ (µm) II. | | $\Delta\sigma_{Rt}$ (%) |
|---|---|---|---|---|---|---|
| | Turned | Burnished | | Turned | Burnished | |
| 1 | 7.4429 | 3.4059 | 54.24 | 7.6052 | 3.8576 | 49.28 |
| 2 | 6.6607 | 9.3875 | −40.94 | 7.8737 | 2.3713 | 69.88 |
| 3 | 7.1770 | 3.6195 | 49.57 | 8.5543 | 3.2045 | 62.54 |
| 4 | 6.7281 | 5.5299 | 17.81 | 9.5248 | 2.5208 | 73.53 |
| 5 | 6.9308 | 16.2023 | −133.77 | 7.1352 | 3.7538 | 47.39 |
| 6 | 6.7881 | 15.7811 | −132.48 | 7.9379 | 3.4761 | 56.21 |
| 7 | 7.1138 | 3.0610 | 56.97 | 8.5208 | 4.8280 | 43.34 |
| 8 | 6.2375 | 9.7202 | −55.83 | 7.5960 | 9.2197 | −21.38 |

The calculated ratios were ranked in order to make it clearer which parameter settings are most advantageous for the improvement of surface roughness. The greatest improvement is marked by number 1, the worst is marked by number 8. Next, these ranks were summarized (R1 + R2 + R3 = Σ); thus, an overall ranking can be established (Tables 8 and 9).

**Table 8.** Ranking of results for Experiment I.

| No. | $\Delta\sigma_{Ra}$ (%) | R1 | $\Delta\sigma_{Rq}$ (%) | R2 | $\Delta\sigma_{Rz}$ (%) | R3 | $\Delta\sigma_{Rt}$ (%) | R4 | Σ | R |
|---|---|---|---|---|---|---|---|---|---|---|
| 1 | 71.80 | 2 | 59.29 | 3 | 55.14 | 3 | 54.24 | 2 | 10 | 2 |
| 2 | −37.69 | 6 | −36.37 | 6 | −17.93 | 5 | −40.94 | 5 | 22 | 4 |
| 3 | 63.82 | 3 | 63.67 | 2 | 57.70 | 2 | 49.57 | 3 | 10 | 2 |
| 4 | 44.99 | 4 | 42.32 | 4 | 31.53 | 4 | 17.81 | 4 | 16 | 3 |
| 5 | −80.06 | 7 | −91.36 | 7 | −52.54 | 7 | −133.77 | 8 | 29 | 5 |
| 6 | −109.46 | 8 | −114.65 | 8 | −69.71 | 8 | −132.48 | 7 | 31 | 6 |
| 7 | 73.38 | 1 | 73.05 | 1 | 66.80 | 1 | 56.97 | 1 | 4 | 1 |
| 8 | −28.64 | 5 | −30.33 | 5 | −21.18 | 6 | −55.83 | 6 | 22 | 4 |

**Table 9.** Ranking of results for Experiment II.

| No. | $\Delta\sigma_{Ra}$ (%) | R1 | $\Delta\sigma_{Rq}$ (%) | R2 | $\Delta\sigma_{Rz}$ (%) | R3 | $\Delta\sigma_{Rt}$ (%) | R4 | Σ | R |
|---|---|---|---|---|---|---|---|---|---|---|
| 1 | 58.18 | 5 | 58.59 | 5 | 55.59 | 5 | 49.28 | 5 | 20 | 5 |
| 2 | 71.71 | 1 | 71.38 | 1 | 66.91 | 2 | 69.88 | 2 | 6 | 1 |
| 3 | 67.57 | 2 | 67.17 | 2 | 63.11 | 3 | 62.54 | 3 | 10 | 3 |
| 4 | 65.85 | 3 | 66.96 | 3 | 69.01 | 1 | 73.53 | 1 | 8 | 2 |
| 5 | 48.65 | 6 | 47.71 | 6 | 50.75 | 6 | 47.39 | 6 | 24 | 6 |
| 6 | 63.42 | 4 | 63.32 | 4 | 61.63 | 4 | 56.21 | 4 | 16 | 4 |
| 7 | 39.19 | 7 | 38.55 | 7 | 37.22 | 7 | 43.34 | 7 | 28 | 7 |
| 8 | −29.44 | 8 | −28.53 | 8 | −21.11 | 8 | −21.38 | 8 | 32 | 8 |

It was found that applying $F_1 = 10$ N burnishing force with $f_2 = 0.005$ mm/rev feed rate and $i_2 = 3$ number of passes (case of No. 7) produces the most preferred surface-roughness values for Experiment I.

In the second experiment, the burnishing speed was also examined, and its lower value $v_1 = 15$ m/min proved to be more advantageous with $F_2 = 20$ N force and $f_1 = 0.001$ mm/rev feed rate (case of No. 2).

The full factorial experimental design method was applied to create experimental formulas (2)–(9) and calculations and axonometric figures (Figures 3–6) were prepared using MathCAD 15 software.

$$\Delta\rho_{Ra1} = 362.513 - 19.261 \cdot F - 6.525 \cdot 10^4 \cdot f - 156.563 \cdot i + \\ +4.307 \cdot 10^3 \cdot F \cdot f + 6.045 \cdot F \cdot i + 4.059 \cdot f \cdot i - 2.041 \cdot 10^3 \cdot F \cdot f \cdot i \tag{2}$$

$$\Delta\rho_{Ra2} = 61.585 - 0.094 \cdot F - 6.165 \cdot 10^3 \cdot f - 1.54 \cdot v + \\ +1.323 \cdot 10^3 \cdot F \cdot f + 0.122 \cdot F \cdot v + 821.667 \cdot f \cdot v - 113.583 \cdot F \cdot f \cdot v \tag{3}$$

$$\Delta\rho_{Rq1} = 323.246 - 16.972 \cdot F - 5.679 \cdot 10^4 \cdot f - 150.814 \cdot i + \\ +3.788 \cdot 10^3 \cdot F \cdot f + 5.549 \cdot F \cdot i + 3.93 \cdot f \cdot i - 1.93 \cdot 10^3 \cdot F \cdot f \cdot i \tag{4}$$

$$\Delta\rho_{Rq2} = 67.093 - 0.42 \cdot F - 7.593 \cdot 10^3 \cdot f - 1.779 \cdot v + \\ +1.417 \cdot 10^3 \cdot F \cdot f + 0.135 \cdot F \cdot v + 865.833 \cdot f \cdot v - 116.15 \cdot F \cdot f \cdot v \tag{5}$$

$$\Delta\rho_{Rz1} = 250.396 - 12.746 \cdot F - 4.04 \cdot 10^4 \cdot f - 111.101 \cdot i + \\ +2.644 \cdot F \cdot f + 4.266 \cdot F \cdot i + 2.931 \cdot f \cdot i - 1.471 \cdot 10^3 \cdot F \cdot f \cdot i \tag{6}$$

$$\Delta\rho_{Rz2} = 56.12 - 0.283 \cdot F - 7.45 \cdot 10^3 \cdot f - 1.006 \cdot v + \\ +1.459 \cdot 10^3 \cdot F \cdot f + 0.103 \cdot F \cdot v + 712.333 \cdot f \cdot v - 106.317 \cdot F \cdot f \cdot v \tag{7}$$

$$\Delta\rho_{Rt1} = 355.298 - 18.146 \cdot F - 6.364 \cdot 10^4 \cdot f - 188.855 \cdot i + \\ +3.804 \cdot 10^3 \cdot F \cdot f + 7.042 \cdot F \cdot i + 4.662 \cdot 10^4 \cdot f \cdot i - 2.219 \cdot 10^3 \cdot F \cdot f \cdot i \tag{8}$$

$$\Delta\rho_{Rt2} = 24.728 + 1.88 \cdot F - 5.938 \cdot 10^3 \cdot f - 0.118 \cdot v + \\ +1.358 \cdot 10^3 \cdot F \cdot f + 0.028 \cdot F \cdot v + 777.0 \cdot f \cdot v - 106.55 \cdot F \cdot f \cdot v \tag{9}$$

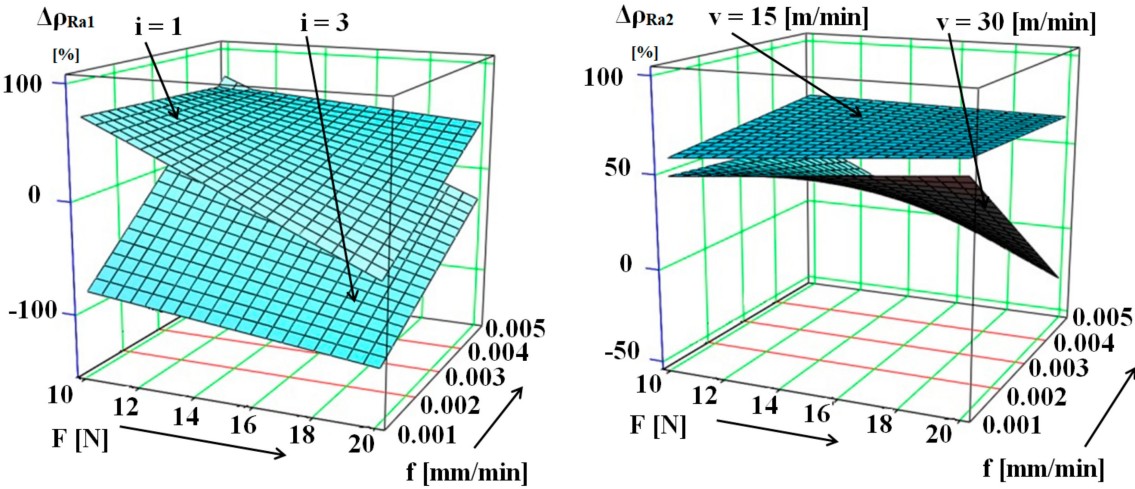

**Figure 3.** Changes in $R_a$ for Experiment I (**left**) and Experiment II (**right**).

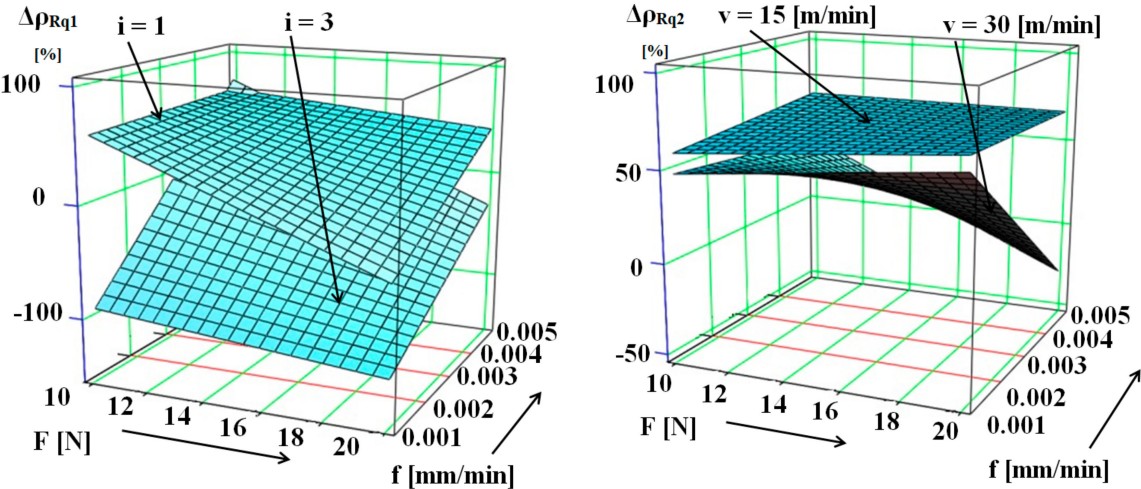

**Figure 4.** Changes in $R_q$ for Experiment I (**left**) and Experiment II (**right**).

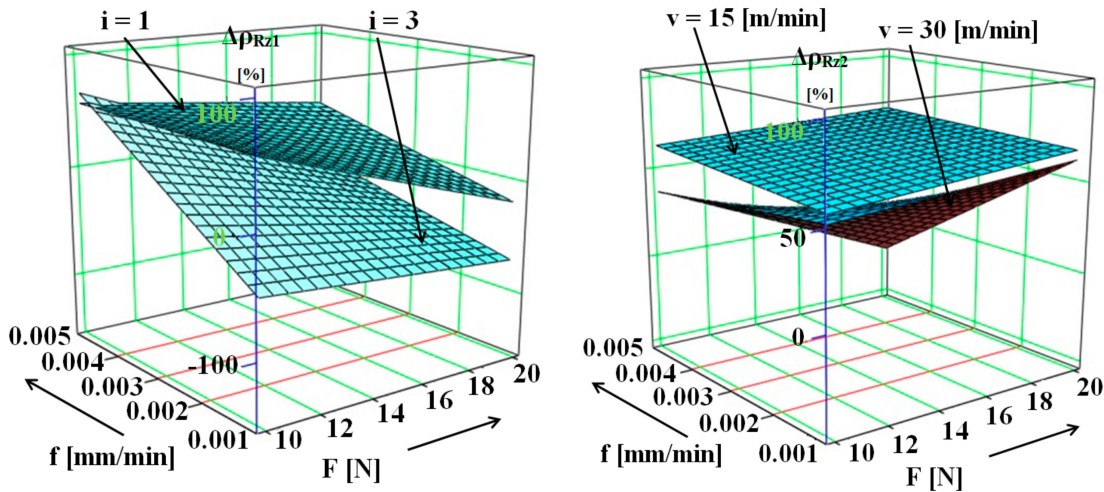

**Figure 5.** Changes in $R_z$ for Experiment I (**left**) and Experiment II (**right**).

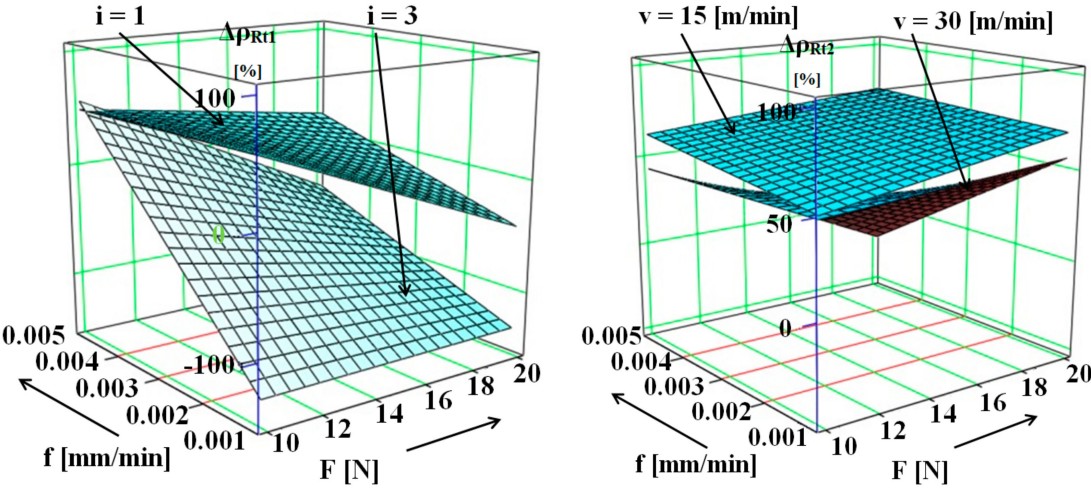

**Figure 6.** Changes in $R_t$ for Experiment I (**left**) and Experiment II (**right**).

## 4. Discussion

This paper presents our experimental investigations on burnishing of low-alloyed aluminium shafts, in which the examined parameters were the burnishing force, feed rate, speed and number of passes. The purpose of the studies was to examine the influence of these burnishing setting parameters on different surface-roughness parameters, such as $R_a$, $R_q$, $R_z$ and $R_t$. The full factorial experimental design method was applied to examine the changes caused by burnishing and to make it even more vivid, dimensionless ratios were used when creating empirical formulas and 3D diagrams were created for each roughness parameter.

According to the measured and calculated results of the two experiments performed, the following statements can be made:

- On the base of main and cross-effect analysis, the following can be stated: In Experiment I, for both $i_1 = 1$ and $i_2 = 3$ number of passes, increasing the burnishing force from $F_1 = 10$ N to $F_2 = 20$ N had a negative effect on the numerical value of the surface-roughness-improvement ratio for all four characteristics ($\Delta\sigma_{Ra}$, $\Delta\sigma_{Rq}$, $\Delta\sigma_{Rz}$ and $\Delta\sigma_{Rt}$).
- Increasing the feed from $f_1 = 0.001$ mm/rev to $f_2 = 0.005$ mm/rev for the number of passes $i_2 = 3$ had a clear positive effect on the value of the surface-roughness-improvement ratio both when applying the burnishing force $F_1 = 10$ N and $F_2 = 20$ N. In contrast to this, in the case of the realization of the number of burnishing passes $i_1 = 1$, a decrease in the value of the surface-roughness-improvement ratio can be discovered.
- In the case of Experiment II, increasing the burnishing force from $F_1 = 10$ N to $F_2 = 20$ N at $v_1 = 15$ m/min burnishing speed showed a positive trend in the values of all the four ($\Delta\sigma_{Ra}$, $\Delta\sigma_{Rq}$, $\Delta\sigma_{Rz}$ and $\Delta\sigma_{Rt}$) surface-roughness-improvement ratios for the low feed ($f_1 = 0.001$ mm/rev).
- At the higher speed ($v_2 = 30$ m/min), increasing the feed from $f_1 = 0.001$ mm/rev to $f_2 = 0.005$ mm/rev, when $F_2 = 20$ N was used, had a negative effect on the tested surface-roughness-improvement ratios. Therefore, the application of a higher burnishing speed ($v_2 = 30$ m/min) and a lower burnishing force ($F_1 = 10$ N) is more beneficial in terms of surface-roughness improvement.
- Following Tables 8 and 9, the beneficial burnishing parameter settings are summarized in in Table 10.

**Table 10.** The most appropriate settings.

| Experiment I | Experiment II |
|---|---|
| $F_1 = 10$ N | $F_2 = 20$ N |
| $f_2 = 0.005$ mm/rev | $f_1 = 0.001$ mm/rev |
| $v_2 = 30$ m/min | $v_1 = 15$ m/min |
| $i_2 = 3$ | $i_1 = 1$ |

- In the future, we intend to examine the effect of the increased number of passes, and we intend to study the 3D roughness parameters to better understand the processes taking place during machining.

**Author Contributions:** Conceptualization: V.F. and G.V.; methodology: V.F. and G.V.; investigation: V.F. and G.V.; writing—original draft preparation: V.F. and G.V.; writing—review and editing: V.F. and G.V. All authors have read and agreed to the published version of the manuscript.

**Funding:** This research was funded by National Research, Development, and Innovation Fund of Hungary, financed under the K_17 funding scheme, grant number NKFI-125117.

**Institutional Review Board Statement:** Not applicable.

**Informed Consent Statement:** Not applicable.

**Data Availability Statement:** Not applicable.

**Acknowledgments:** Project no. NKFI-125117 has been implemented with the support provided from the National Research, Development and Innovation Fund of Hungary, financed under the K_17 funding scheme.

**Conflicts of Interest:** The authors declare no conflict of interest.

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
