# Peer review of "The Influence of Diamond Burnishing Process Parameters on Surface Roughness of Low-Alloyed Aluminium Workpieces"

_machines, doi:10.3390/machines10070564_

Round 1

Reviewer 1 Report

I recommend authors that English and style require fine/minor spelling checks.

Author Response

The style and spelling was checked by a native lector.

Reviewer 2 Report

The authors have taken good research work; however, the authors need to carry out the following corrections before the publication of this work. 

1. Explain all symbols used in the article.

2. Give the full name of devices used in research.

3. In the discussion, the authors mention that, "burnishing force is strongly correlated with the number of passes", but on what basis? Please explain.

4. Conclusions should contain plans for further research.

Author Response

The authors have taken good research work; however, the authors need to carry out the following corrections before the publication of this work.

Point 1. Explain all symbols used in the article.

Response 1: All symbols used in the article were explained.

Point 2. Give the full name of devices used in research.

Response 2: We used:

  • OPTIMUM type OPTIturn S600 CNC lathe
  • Altisurf 520 measuring machine with CL2 confocal chromatic sensor.

The paper containes these full denominations.

Point 3. In the discussion, the authors mention that, "burnishing force is strongly correlated with the number of passes", but on what basis? Please explain.

Response 3: You are right!

We rewrote the whole discussion.

According to the measured and calculated results of the two experiments performed, the following statements can be made:

  • On the base of main and cross effect analysis it can be stated as follows: In Experiment I, for both i1=1 and i2=3 number of passes, increasing the burnishing force from F1=10 N to F2=20 N had a negative effect on the numerical value of the surface roughness improvement ratio for all four characteristics (ΔσRa, ΔσRq, ΔσRz and ΔσRt).
  • Increasing the feed from f1=0.001 mm/rev to f2=0.005 mm/rev for the number of passes i2=3 had a clear positive effect on the value of the surface roughness improvement ratio both when applying the burnishing force F1=10 N and F2=20 N. In contrast to this, in the case of the realization of the number of burnishing passes i1=1, a decrease in the value of the surface roughness improvement ratio can be discovered.
  • In the case of Experiment II, increasing the burnishing force from F1=10 N to F2=20 N at v1=15 m/min burnishing speed showed a positive trend in the values of all the four (ΔσRa, ΔσRq, ΔσRz and ΔσRt) surface roughness improvement ratios for the low feed (f1=0.001 mm/rev).
  • At the higher speed (v2=30 m/min), increasing the feed from f1=0.001 mm/rev to f2=0.005 mm/rev, when F2=20 N was used, had a negative effect on the tested surface roughness improvement ratios. Therefore, the application of a higher burnishing speed (v2=30 m/min) and a lower burnishing force (F1=10 N) is more beneficial in terms of surface roughness improvement.
  • In pursuance of Tables 8 and 9, the beneficial burnishing parameter settings are summarized in in Table 10.

Table 10. The most appropriate settings

Experiment I

Experiment II

F1=10 N

F2=20 N

f2=0.005 mm/rev

f1=0.001 mm/rev

v2=30 m/min

v1=15 m/min

i2=3

i1=1

Point 4. 4. Conclusions should contain plans for further research.

Now it can be found in the last sentence:

  • In the future we intend to examine the effect of the increased number of passes, and we intend to study the 3D roughness parameters to better understand the processes taking place during machining.

Reviewer 3 Report

A very interesting article.

Author Response

(The authors gave the same response as above.)
